# Tumor-Associated Macrophages in Osteosarcoma: From Mechanisms to Therapy

**DOI:** 10.3390/ijms21155207

**Published:** 2020-07-23

**Authors:** Francesca Cersosimo, Silvia Lonardi, Giulia Bernardini, Brian Telfer, Giulio Eugenio Mandelli, Annalisa Santucci, William Vermi, Emanuele Giurisato

**Affiliations:** 1Department of Biotechnology Chemistry and Pharmacy, University of Siena, 53100 Siena, Italy; francesca.cersosi@student.unisi.it (F.C.); giulia.bernardini@unisi.it (G.B.); annalisa.santucci@unisi.it (A.S.); 2Department of Molecular and Translational Medicine, University of Brescia, 25123 Brescia, Italy; silvia.lona@gmail.com (S.L.); g.mandelli@unibs.it (G.E.M.); william.vermi@unibs.it (W.V.); 3Division of Pharmacy and Optometry, School of Health Sciences, Faculty of Biology, Medicine and Health, The University of Manchester, Manchester M13 9PL, UK; Brian.Telfer@manchester.ac.uk; 4Department of Pathology and Immunology, Washington University School of Medicine, St. Louis, MO 63110, USA; 5Division of Cancer Sciences, School of Medical Sciences, Faculty of Biology, Medicine and Health, The University of Manchester, Manchester M13 9PL, UK

**Keywords:** tumor-associated macrophages (TAMs), osteosarcomas (OSs), metastasis, cancer-stem cells, mesenchymal stem cells, tumor microenvironment, therapy

## Abstract

Osteosarcomas (OSs) are bone tumors most commonly found in pediatric and adolescent patients characterized by high risk of metastatic progression and recurrence after therapy. Effective therapeutic management of this disease still remains elusive as evidenced by poor patient survival rates. To achieve a more effective therapeutic management regimen, and hence patient survival, there is a need to identify more focused targeted therapies for OSs treatment in the clinical setting. The role of the OS tumor stroma microenvironment plays a significant part in the development and dissemination of this disease. Important components, and hence potential targets for treatment, are the tumor-infiltrating macrophages that are known to orchestrate many aspects of OS stromal signaling and disease progression. In particular, increased infiltration of M2-like tumor-associated macrophages (TAMs) has been associated with OS metastasis and poor patient prognosis despite currently used aggressive therapies regimens. This review aims to provide a summary update of current macrophage-centered knowledge and to discuss the possible roles that macrophages play in the process of OS metastasis development focusing on the potential influence of stromal cross-talk signaling between TAMs, cancer-stem cells and additional OSs tumoral microenvironment factors.

## 1. Introduction

Osteosarcoma (OS) is the most common type of malignant primary tumor in bone tissue, affecting mainly children and young adults. Metastatic progression to pulmonary tissue and subsequent patient relapse remains the leading cause of mortality in the clinical setting for OS patients [1]. Current treatments for OS patients typically include surgery in combination with adjuvant and neo-adjuvant chemotherapeutic agents. The survival rate for metastatic OS patients is currently estimated at less than five years [2]. The existence of high levels of cellular heterogeneity, and the complexity of the molecular and genetic mechanisms associated with osteosarcomagenesis makes it extremely difficult to develop a singular therapeutic approach in the clinical setting.

In recent years, research into the molecular cell signaling components involved in OS has led to an interest in evaluating potential therapeutic targets to block disease progression. Communication between OS cells and the surrounding tumor microenvironment (TME) is necessary for tumor growth and subsequent metastases. A major influential component within the TME are tumor-associated macrophages (TAMs) that are immune cells involved in inflammatory responses and tissue homeostasis [3]. In most solid cancers, increased infiltration of TAMs has long been associated with poor patient prognosis highlighting their value as potential diagnostic and prognostic biomarkers in cancerous tissues. Recently, heterogeneous population of TAMs with an intermediate phenotype M1–M2 have been observed in primary OS associated with anti-metastatic activity [4]. However, a number of studies have also indicated a role for TAMs in metastatic OS. A higher density of M2-type TAMs were found in lung metastases compared to primary OS in association with proinflammatory molecules resulting in increased tumor invasive capability [5]. Additional studies demonstrated the participation of TAMs in promoting angiogenesis via the activation of intracellular signaling pathways known to be involved in cancer progression [6,7]. In addition, blocking M2-polarization was correlated with anti-metastatic activity in OS and reduced expression of stem cell-like properties [8].

In this review, we aim to provide a current and comprehensive update about the known factors participating in the recruitment and activation of TAMs and the mechanism by which TAMs supports tumor metastasis in OS, focusing attention on the relationships between the signaling regulating the cross-talk between TAMs, cancer stem-cells (CSCs) and mesenchymal stem cells (MSCs) in primary and metastatic OS. Although anticancer therapy based on macrophage-centered approaches have been investigated, a better understanding of the role of TAMs in OS is required to improve and develop therapeutic strategies targeting macrophages.

## 2. Osteosarcoma and Macrophage Function

### 2.1. Osteosarcoma: Clinical and Molecular Features

Osteosarcoma is a primary malignant bone tumor mainly affecting children and young adults in the age range between 10–20 years old [1]. With a worldwide incidence of 3/100,000, OS commonly develops in the metaphyseal growth plates of the elongating long bones. Typical signs and symptoms of OS are localized pain followed by localized swelling and limited joint movement, while pathologic fracture at the site of the disease is a rare event [9]. OS is a highly metastatic cancer. OS metastasis preferentially localizes in lung tissue with distant bones and lymph nodes as the second most common locations sites. Metastatic disease and relapse remain the leading cause of mortality for OS patients. Around 30% of OS patients with localized disease and around 80% with metastatic disease at diagnosis will relapse [1]. Despite current aggressive therapy approaches, prognosis for OS patients remains poor with less than a 30% survival rate after five years for patients presenting metastatic disease or patients who have relapsed post treatment. In contrast, the survival rate for patients with localized OS is around 75% [2]. Histological analysis of localized OS in bone tissue shows a measurable high-grade neoplasm within the medullary as the most frequent indicative factor observed [10]. OS can be further sub-classified or graded depending on the location of the neoplasm in relation to the medullary tissue and can be defined as osteoblastic, chondroblastic and fibroblastic OS [10]. Rare variants of OS are based on distinct intrinsic morphologic patterns including telangiectatic OS, small cell OS and low-grade central OS. Additional classifications can be defined using the site of the disease and are denoted as parosteal OS, periosteal OS and high-grade surface OS. These localized variants can show distinctive clinical characteristics [11,12]. OS originates from oncogenic and epigenetic events involving primitive mesenchymal cells (MSCs) which, once differentiated into osteoblasts, produce malignant osteoid and immature bone tissue [13]. Most OS occurs sporadically and is characterized by high heterogeneity both at the intra- and inter-tumoral level. OS possesses unique features compared to other pediatric cancers, including an extremely complex genomic background with a high degree of genomic instability and has an unusually high mutation rate and a high number of mutated driver genes [14]. In a small percentage of patients, OS has a genetic origin. The association between the increased risk of developing OS and several rare inherited cancers syndromes, such as Li–Fraumeni and retinoblastoma familial cancers (germ-line mutations in *TP53* and *RB1*, respectively) or Rothmund–Thomson, Werner and Bloom (germline mutations in DNA helicase genes *RECQL4, RECQL2* and *RECQL3*, respectively), have been well documented [15]. Recently, other germ-line pathogenic variants have been detected in genes not previously associated with OS such as *CDKN2A*, *MEN1*, *VHL, POT1, APC, MSH2* and *ATRX* in more than 25% of OS patients [16]. High-grade OS is characterized by widespread genomic alterations encompassing hyperdiploidy, chromosomal re-arrangements, copy number variation and a very high somatic mutation rate [14,17]. Recurrent mutations, in a set of cancer-related genes, have been identified by genome-wide NGS techniques [18,19]. Alteration of onco- or tumor-suppressor genes that are involved in signaling pathways regulating three core cellular processes: cell fate, cell survival, and maintenance of genome stability have also been detected [19]. Loss-of-function variants have been observed for *TP53*, *RB1*, *DLG2*, *PTEN*, *CDKN2A*, *ATRX* genes and are mainly caused by deletion, mutation or translocation events or by copy number variations; while gain-of function variants have been observed for *MYC*, *CCNE1*, *COPS3*, *AURKB*, *CDK4*, *MDM2*, *IGF1R* and *AKT* genes and are mainly due to gene amplification events. Among these, TP53 and RB1 tumor suppressor genes have shown the highest mutation frequency [20]. Loss of function mutations for these two have been reported as the main drivers of the events resulting in the development of sporadic OS [20]. Furthermore, mutations of several regulators of *p53* and *RB1* activity were also observed including amplification of the *MDM2* and *CDK4* genes. A whole exome sequencing investigation uncovered mutations in alternative driver genes, including a *BRCA1/2* defect [18]. OS cells also showed alterations in their epigenetic machinery including DNA methylation, histone modifications and ncRNAs expression [21]. Functional analysis of the differentially methylated genes revealed that these genes are mainly involved in biologic processes related to an inflammatory/immune response involving pertussis pathways and hematopoietic cell lineage pathways [22]. Additionally, hypomethylation of oncogenic elements have also been observed [23,24]. Epigenetic alterations have been associated with resistance to chemotherapy and relapsing forms of OS [25]. Aberrant expression of both onco- and tumor-suppressor genes, generated by genetic and epigenetic events, results in uncontrolled cell-signaling pathway and cellular processes. The PI3K/Akt pathway is universally accepted as one of the most important oncogenic drivers in OS [26,27]. The Wnt/ β-catenin-signaling pathway is markedly upregulated in OS and plays a pivotal role in tumorigenesis [28,29] and metastases [30]. Recently, the involvement of this pathway in angiogenesis and immuno-surveillance has been also reported [31,32]. Increasingly data suggest the abnormal activation of the hedgehog (Hh) pathway that has a pivotal role in signaling during embryogenesis, in OS [33,34] and several therapeutic strategies have been developed to target this pathway [35,36]. The prognostic and predictive impact of the expression of *p16* [37], *HER2* [38] and *β-catenin* [39] have also been investigated with some promising results.

Tumorigenesis and cancer progression is supported by crosstalk-signaling between OS cells and the surrounding TME, being mainly composed of stromal cells, immune cells, endothelial cells and bone cells [40]. The immune component of OS TME is predominantly represented by the presence of tumor-associated macrophages (TAMs) that may act to impede or enhance the development and progression of tumor (Figure 1).

### 2.2. Macrophages and Bone Homeostasis

Macrophages are a heterogeneous population of innate myeloid cells that can infiltrate all mammalian tissues [42] and have multiple roles in organ development, immune surveillance, homeostasis and tissue turnover [43]. Expressing a broad array of cell-surface receptors, intracellular mediators and secretory molecules, macrophages play an essential role during the inflammatory response and coordinate host defense mechanisms [44,45].

Macrophages are involved in bone homeostasis and immunity and have central functions in osteoimmunology [46]. Three known distinct macrophage populations have been identified in the bone skeletal tissue: bone marrow macrophages (erythroid island macrophages and hematopoietic stem cell macrophages), osteoclasts (multinucleated tissue-resident macrophages in bone) and a population of macrophages termed osteal macrophages or “osteomacs” [47]. The role of macrophages in bone biology has been characterized in vitro and in vivo. Human and murine macrophages produce the bone morphogenetic proteins-2 (BMP-2) and -6 (BMP-6) [48] required for mineralization and osteoblastic differentiation gene expression [47]. Interestingly, human macrophages-derived oncostatin M (OSM) enhances MSCs differentiation into osteoblasts [49,50] and matrix mineralization [51]. In addition, it has also been reported that the number of osteoblasts lining bone surfaces and bone formation capability decreases in macrophage ablation models and that osteoporosis increases in skeletally mature mice [52,53,54,55,56].

Macrophages possess high plasticity and can acquire opposing phenotypes: the inflammatory phenotype or classical (M1) and the anti-inflammatory or alternative phenotype (M2) [57,58,59]. Polarized macrophages have different functional roles in bone homeostasis. In studies, it was found that the expression of the osteoinductive factors BMP-2 and BMP-6 were reduced in M1-like macrophages in the presence of M1-conditioned medium and the differentiation of MSCs into osteoblasts was subsequently impaired [48]. Whereas M2-like macrophages participate in the efferocytosis (clearance of apoptotic cells) [60] and conditioned media from M2-like macrophages increased osteoblast maturation from MSCs [50], indicating that alternatively activated macrophages work as mediators of bone homeostasis.

### 2.3. Macrophages: Origin and Plasticity

Previous reports looking at tissue macrophage derivation, suggested that they were derived from circulating monocytes that differentiate into mature macrophages once they are recruited into tissues [61,62]. However, emerging evidence suggests that macrophages in many steady-state tissues are not derived from circulating monocytes as originally thought, but rather from embryonic macrophages that are able to self-maintain independently of the adult hematopoietic system [63,64,65,66,67,68]. Based on more recent work it is now thought that macrophages originate from at least three different embryonic sources; from macrophage/dendritic cell progenitor cells (MDPs) in the bone marrow; from erythro-myeloid progenitor (EMP) in the yolk sac and fetal liver) and differentiate into tissue-specific macrophages; or differentiate into other cell types (such as common dendritic cell progenitors (CDP) [69,70]. The different origin of macrophages is regulated by different regulatory mechanisms; for example, bone-marrow derived cells require transcription factor cMYB, which is not required for yolk-sac derived macrophages [63,71,72,73]. Nevertheless, regardless of their origins, colony-stimulating factor 1 receptor (CSF1R) has been identified as the principal transmembrane receptor that controls differentiation and survival for most macrophages [74].

Activation of macrophages is usually initiated via responses to stimuli present in their immediate surrounding microenvironment [57,75,76]. They possess high plasticity and, according to signals derived from the microenvironment, macrophages can adopt different functional phenotypes during the different phases of an inflammatory reaction [57]. Classical activation, by Toll-like receptors and interferon-γ (IFNγ), gives rise to M1 macrophages that play key roles in the hosts defense against infections, tumor suppression and immune system stimulation. In addition, M1 macrophages are characterized by the production of inducible nitric-oxide synthase (iNOS) and the expression of cytokines involved in the induction of T helper type-1 (Th1) cell development, T-cell trafficking as well as chemo-attractants [77]. On the other hand, alternative activation—by interleukins-4 or -13—induces M2 macrophages that are involved in immune suppression, tumor angiogenesis, tissue remodeling and tumor progression [78,79]. However, it is becoming clear that M1 and M2 classification is possibly an oversimplification of the complexity of macrophage-polarization. In fact, due in part to their plasticity, macrophages exist in different functional states and can be re-directed from one phenotype to another by microenvironmental changes. In particular, M2 macrophages consist of a spectrum of phenotypes identified as M2a, M2b, M2c and M2d [80,81], and the original two designations are currently referred to as ‘M1-like’ and ‘M2-like’, due to the overlap of the expression of markers. TME induces the M2-polarization of tumor-associated macrophages (TAMs) and a high M2/M1 ratio is clinically associated with poor prognosis in many types of cancers [82].

Tumor-associated macrophages are important components of the tumor stroma and are intimately involved in many phases of tumor growth. In several cases, macrophages can comprise up to 50% of the tumor mass and their abundance is associated with a poor clinical outcome. A great deal of evidence has demonstrated that TAMs promote tumor growth by facilitating angiogenesis, immunosuppression and chronic inflammation and can also influence tumor resistance after conventional anticancer therapies [59]. As mentioned earlier, it has been previously hypothesized that TAMs originate from circulating monocytes. However, new evidence suggests that, while monocyte-derived TAMs are continuously replenished by peripheral recruitment, a proportion of TAMs can also arise from tissue-resident macrophages that are locally maintained through in situ proliferation [83,84]. TAMs play three distinct roles in promoting tumor growth and metastases: first; TAMs facilitate the intravasation of tumor cells into the vasculature via a paracrine-signaling loop consisting of macrophages CSF-1 from the tumor cells and EGF from macrophages and their receptors [85,86], thereby promoting tumor dissemination [87]. Indeed, VEGF-A-expressing and Tie2-expressing macrophages produce highly localized vascular permeability thereby facilitating the extravasation of the tumor cells [88,89]; second, TAMs promote tumor growth by inhibiting both adaptive and innate antitumor immunity by secreting immunosuppressive molecules including TGFβ, IL10, arginase-1 (Arg-1) and NO [90,91,92,93]; third, TAMs are proangiogenic, thereby promoting tumor growth and recovery from cancer therapy. Notably, it has been found that VEGF-A-expressing and Tie2-expressing macrophages play a crucial role in the recovery of the tumor vasculature and the recurrence of tumor following treatment with doxorubicin [94,95].

## 3. TAMs and TME in OS

### 3.1. TAMs and OS Metastasis

Initial findings, providing evidence of TAMs in OS, showed that neoplastic tissue was being infiltrated with heterogeneous populations of TAMs with M1 and M2 phenotypes [4]. Gene profiling analysis of nonmetastatic OS samples revealed a high expression of macrophage-associated genes and a high number of macrophages responders to chemotherapy treatment [4]. Despite the introduction of neoadjuvant chemotherapy increasing the overall survival of OS patients, the progression to OS metastasis has still not been overcome in the clinic. Liang et al. showed that neoadjuvant chemotherapy induced severe local inflammation responses in OS patients resulting in chemotherapy-induced infiltrating macrophages to release interleukin-1-beta (IL-1β) and subsequent activation of the TNF, NF-kB and MAPKs pathways. Indeed, the binding of IL-1β to its receptor, on the OS cells, resulted in reduced sensitivity to neoadjuvant chemotherapy [96]. Previous research findings demonstrated the presence of reduced tumor infiltrating T cells (Tis) correlated with the persistence of M2-type TAMs in OS tumors. Furthermore, the depletion of M2-types restored T cell proliferation and T cell secretion of TNF-α and INF-β. Interestingly, the levels of CD163^+^ TAMs were higher in the tumor stroma compared with the frequency of circulating monocytes in peripheral blood, suggesting that the regulation of the T cell response in OS is not a systemic condition, but related to factors in TME [97].

With regard to the role of TAMs in the metastatic process, Dumars et al. showed high levels of CD146^+^ vascular cells in metastatic OS samples correlating with the presence of CD163^+^ TAMs, compared with nonmetastatic cells, suggesting a role for M2-types in the promotion of the neoangiogenic process and thereby enhancing the metastatic potential of OS cells [6]. In vitro and in vivo studies showed that metastatic OS tissues overexpress IL-34 that is involved in monocyte differentiation and inflammation. When OS cells were stimulated with the proinflammatory cytokines, TNF-α and IL-1β, a higher expression of IL-34 was reported [7]. Additionally, IL-34 was found to promote vasculogenesis and OS growth via activation of signaling pathways in endothelial cells including Src, FAK and MAPKs, known to be involved in angiogenesis and vascular homeostasis. Moreover, in vivo data showed increases in the numbers of M2-macrophages in IL-34 expressing tissues [7]. IL-34 strongly induced the adhesion of monocytes to endothelial cells thereby enhancing the extravasation capability of TAMs in OS [7]. Su Y et al. also reported overexpression of CCL18, a common chemokine released by M2-macrophages, in OS tissues. CCL18 expression was correlated with proliferation and invasion of OS cells. Additionally, the number of CCL18+ TAMs identified was higher in metastatic OS tissues compared with primary OS. Studies in xenograft models indicated that CCL18 increased the tumor size and induced lung metastasis, indicating that TAMs, by secreting CCL18 promote metastasis and tumor growth in OS [98]. Higher densities of TAMs in metastatic OS tissues has been recently recorded correlating with a marked increase in the expression levels of COX2 and p-STAT3 (Table 1).

While the inhibition of COX2 reduced the migratory ability of OS cells, COX2 overexpression in OS cells co-cultured with TAMs increased expression levels of p-STAT3 and thus the metastatic potential of OS cells [5]. The pro-metastatic role of TAMs has already been proposed by Zhou et al. reporting that treatment with all-trans retinoic acid (ATRA) not only prevented M2-polarization, but also showed anti-metastatic activity in OS by blocking TAM-induced MMP12 secretion [8]. Moreover, Shao X et al. observed a higher percentage of CD209^+^ M2-TAMs in primary OS. Additionally, inhibition by ATRA reduced tumorigenesis in OS by blocking M2-polarization and, more interestingly, suppressed the stem-like properties of OS cells such as the colony formation capability, reduced the number of CD117^+^/Stro-1^+^ cells and the expression of stemness genes [99].

### 3.2. Crosstalk between TAMs, CSCs and MSCs in OS

Several tumors, including breast, lung and colorectal cancers, have been found to contain stem-like cell populations termed cancer stem cells (CSCs), with capabilities for tumorigenesis, metastasis and drug resistance [103]. CSCs express high levels of drug transporters and high capability for DNA repair making them more able to resist conventional chemotherapy [104]. In OS, high levels of the detoxifying enzyme aldehyde dehydrogenase-1 (ALDH1) have been associated with drug resistance in stem-like populations [105]. Studies have identified increased levels of stem-like cell gene expression and activation, notably OCT3/4, SOX4 and NANOG, identified in a small fraction of CD133^+^ cells in primary human OS [106]. Tumor-initiating cells (TICs) with stem-like properties have been further identified using the mesenchymal markers CD117 and Stro-1 in human and mouse OS with increased metastatic and drug resistant capabilities [107]. The mesenchymal origin of the tumor and the high expression of mesenchymal markers suggest mesenchymal stem cells (MSCs) as one of the cells-of-origin involved in osteosarcomagenesis [108].

In OS, MSCs are recruited from bone marrow (BM) and, in response to genetic/epigenetic changes and/or micro-environmental signals, differentiate or aberrantly differentiate [109]. Brune et al. identified a subpopulation of OS-MSCs that share molecular and phenotypic characteristics with BM-MSCs, supporting the hypothesis that tumor stromal MSCs can originate from BM-MSCs [110]. Malignant transformation of mouse BM-MSCs was also observed in presence of mutated TP53 and CDKN2A genes [111]. Moreover, TP53-deficient MSCs have been reported as being associated with an increased expression of markers promoting tumorogenesis, immune suppression, drug resistance and metastasis in tumor microenvironment involving CCL5, TGF-β and IL-6/STAT3 [20]. However, the incidence of OS is higher when the alteration of cell-cycle regulators occurs in osteoblast progenitors rather than immature MSCs, suggesting that OS cells-of-origin may be both osteoblast differentiating committed cells and early mesenchymal progenitors, being orchestrated via signals in the tumor microenvironment and/or as a direct response to genetic/epigenetic alterations within the cells-of-origin [108]. Le Nail et al. reported that the injection of OS-derived cells alone was not sufficient to induce tumor initiation in mouse model, while the co-injection of MSCs increased the metastatic potential of OS cells, suggesting the supporting role of MSCs in tumor progression [112]. Many studies have demonstrated that MSCs promote the metastatic potential of tumor cells by enhancing their ‘invasiveness’ as well as their ability to create metastatic niches [113,114,115,116]. MSCs, under TME stimulation, may acquire a ‘cancer-associated fibroblast’ (CAF) phenotype with strong invasive and migratory capabilities within the TME [117]. In addition, overexpression of IL-6 from MSCs has been correlated with the enhanced expression of ICAM-I in tumor stromal cells resulting in the invasive potential of cancer cells to increase [109]. Additional studies have demonstrated the protective role of MSCs from the effects of anticancer treatments in OS [109,118].

In the tumor niche, CSCs communicate with the surrounding stromal components, including immune cells that provide signals supporting the development of the CSCs phenotype and as a result increases the malignant behavior of the tumor. A correlation between TAMs and CSCs has been previously demonstrated in several tumors. TAMs secrete factors promoting the inflammatory response and supporting stem-like phenotype of CSCs [119]. Several molecules have been identified as mediators in crosstalk CSCs–TAMs-signaling, where TAMs were found to induce EGF/STAT3-signaling in breast tumors [120], IL-10/JAK1-signaling in non-small lung cancer cells [121] and TGF-β1 in hepatocellular carcinoma [122]. Inhibiting TAMs resulted in a decreased number of tumor initiating cells and increased the chemoresistance response in pancreatic tumors [123]. More recently, it has been suggested that the involvement of M2-TAMs, in tumor initiation, acts by inducing the stem-like phenotype of the cancer stem cell in OS [99]. However, the TAMs-CSCs crosslink in OS remains to be fully understood.

The paracrine communication between TAMs, MSCs and CSCs in TME plays a key role in supporting the stem-cell niche and tumor development [113]. In this regard, Jia XH et al. reported that MSCs promoted differentiation of macrophages toward M2 phenotype in the tumor via IL-6 secretion [100]. In multiple myeloma, a bidirectional signaling between macrophages and MSCs was reported. Studies have shown that MSCs stimulate the differentiation of macrophages in favor of tumor-supportive phenotypes and that the macrophages in turn promote the pro-tumor activity of MSCs [124]. Another study reported “two-loop signaling” in breast cancer, referring to the interplay between breast cancer cells, that recruit MSCs before the MSCs in turn recruit TAMs via chemokine-signaling [125].

### 3.3. Extracellular Vesicles in OS

Several studies have reported and described the presence of extracellular vesicles (EVs) in the OS microenvironment and their involvement in angiogenesis, tumor progression, immune escape and drug resistance [126]. EVs are small lipid-membrane vesicles released from both normal and tumor cells in the extracellular matrix (ECM) containing proteins, lipids and small nucleic acids able to influence the biologic behaviors of target cells [127]. Cargo including matrix metalloproteinases-1 and -13 (MMP-1, -13), TGF-β, CD-9 and RANKL have been detected in OS-derived EVs [128]. EVs containing TGF-β are able to induce the IL-6-signaling pathway in MSCs (Figure 2), favoring the differentiation into the cancer phenotype [126]. In addition, an immunosuppressive function has been also proposed for OS-derived EVs [129]. Raimondi L. et al. observed that EVs within the OS TME increased the expression of VEGF, IL-8, IL-6 and miRNAs supporting vasculogenesis and tumor dissemination [130]. Tumor-derived EVs have also been reported to participate the development of multidrug resistance phenotypes with regard to sensitive OS cells [131]. Yan B. et al. reported that exosomes secreted by TAMs induced proliferation and invasion of cancer cells and promoted the drug-resistance phenotype by activating the PI3K-AKT-signaling pathways in the OS cells [132] (Figure 2). A recent study reported that the presence of alveolar macrophages in TME induces the secretion of OS-derived EVs [101]. Moreover, they reported that EVs secreted by metastatic OS increased the expression of M2-type markers such as TGFB2, IL10 and CCL22, reducing the phagocytic capability of alveolar macrophages [101]. Although further evaluations are needed to assess the role of TAM-EVs in OS progression, these data identify tumor-associated EVs as potential biomarkers for OS diagnosis which could lead to the development of promising molecular targeted therapies for the effective treatment of OS patients.

## 4. TAM and OS Therapy

Current OS treatment consists of neoadjuvant multidrug chemotherapy comprising cisplatin, doxorubicin, methotrexate (MAP therapy) and Ifosfamide, followed by surgery and postoperative chemotherapy [136]. Radiotherapy is also included used to prevent the growth of the tumor located in high risk areas and as palliative treatment for bone pain in pediatric patients [137]. A five years overall survival rate can be estimated for patients with primary OS by following the aforementioned therapeutic approaches; however, poor therapy response and no improvements in the outcome in patients with metastatic OS has been reported for many years in the clinical setting [136].

Post-surgery histological evaluation has long been used as prognostic tool for OS outcome. Nowadays, new prognostic molecular biomarkers are available, including single nucleotide polymorphism, microRNAs and epigenetic factors that have been useful in identifying the tumor stage, tumor behaviors and the specific therapeutic approach needed to improve the clinical outcome of OS patients [136]. Recently, tumor inflammation was also considered as a prognostic tool for OS, given that stromal inflammation has been correlated with poor prognosis, tumor aggressiveness and metastasis. The use of anti-inflammatory drugs in combination with chemotherapy has also been reported and appears to have a direct effect on increasing patient survival rates [138].

### TAMs as a Target in OS Therapy

Evidence indicates that the ratio of M1 to M2 macrophages may regulate the potential for OSs to metastasize by changing the TME to one that is conducive for metastasis to occur [139]. Therefore, by targeting strategies that modulate TAM-polarization from an M2 phenotype to an M1 phenotype may have a beneficial result in minimizing the effects seen during macrophage aided tumor progression.

Clinical trials have shown that when the liposome-encapsulated muramyl tripeptide mifamurtide (a macrophage activating agent) was used, in combination with standard chemotherapy regimens, resulted in an increase of the 6-year survival rates in OS patients [140,141,142]. In phase II clinical trials mifamurtide induced the infiltration of activated macrophages into OS lung tissue metastases [143] with a marked difference in disease progression and overall survival [144] and that the mifamurtide-mediated M1-TAMs antitumor activity required IFN-γ [145]. Punzo et al. demonstrated that, by switching macrophage polarization towards a TAM-like intermediate M1/M2 phenotype, mifamurtide may work by modulating macrophage functions and inhibit OS proliferation [102]. In order to enhance the activation of the immune response, the European Medical Agency (EMA) has approved the use of mifamurtide, as an adjuvant in combination with chemotherapy, for the treatment of OS patients [146]. In pre-clinical models of OS, the prevalence of M2-type TAMs was correlated with an increase in tumor progression, angiogenesis and metastatic dissemination [6,7,8,147]. It is currently thought that M2-like TAMs, may be suitable targets for anti-metastatic therapy in treating OS patients (Table 2). Recent evidence has demonstrated that All Trans retinoic acid (ATRA) suppresses tumor metastases [148,149], reduces the polarization and infiltration of M2-like TAM cells in the TME and may abrogate pulmonary OS metastasis [8]. M2-like TAM-polarization may also be influenced by the action of the dihydroxycoumarins esculetin and daphnetin, acting via regulation of pSTAT3 and IL-10 that may inhibit the differentiation of M2-types and have reductive effect on OS metastasis in vitro and in vivo [150].

zoledronate (a nitrogen-containing bisphosphonate) also modulates TAM-polarization from M2 to M1 phenotypes [151]. It has also been reported that the TAM functions are affected by bisphosphonates [152]. Drugs such as clodronate and zoledronate are anti-resorptive drugs that have direct effects on osteoclasts and can interrupt the vicious cycle between osteoclasts and bone cancer cells as well as significantly diminishing OS-induced lung metastasis in vivo [153,154,155].

Another approach used in several in vivo research models investigating common types of cancers, is the use of macrophages depletion techniques [156]. Recently, Zhou Q. et al. reported that in clodronate liposome (CLO-LIP) treated mice, M2-like TAMs were actively depleted, and the incidence of lung OS metastasis was markedly reduced [8].

Several studies have shown the importance of macrophage and cancer cell interaction for the intravasation and invasion at distant sites [86,157]. The crosstalk between VCAM1 receptor on the tumor cell surface and α4 monocytes integrins has been shown to be an important factor in several in vivo mouse tumor models [157]. Treatment with natalizumab, a monoclonal antibody against the α4 integrin, has been shown to decrease tumor growth in mouse bone marrow [157]. Currently, natalizumab is in clinical trials for OS (ClinicalTrials.gov Identifier: NCT03811886). Targeting agents that interfere with the signaling crosstalk between cancer cells and TAMs, such as via VCAM-1/α4 integrin, could be an attractive approach to developing novel therapies for more effective OS treatments and improved patient survival rates in the clinical setting.

In addition, it has been reported that program death ligand-1 (PDL-1) and PDL-2 were detectable in human OS cells and OS patient lung metastases [159,160,161,162,163,164]. By targeting the immune checkpoint PD-1/PD-L1-signaling pathway, researchers have reported a reduction in tumor growth and an increase in T-cell immune responses in OS mouse models, indicating possible application as therapeutic targets for future OS treatment [165]. Anti-PD1 antibody led to increased numbers of antitumor M1-like TAMs that correlated closely with reduced M2-like TAM numbers providing a novel mechanism for PD-1 blockade [158]. Clinical trials are currently underway involving the use and evaluation of nivolumab and pembrolizumab (anti-PD1 mAbs) as therapeutic treatments in pediatric OS patients [134]. However, there is still a great deal of research work needed in order to develop effective and targeted therapies for successful OS treatments outcomes in the clinic.

## 5. Future Perspectives

As in many solid tumors, macrophages represent the main immune component in the OS micro-environment and therapies that focus on targeting TAMs have become a hot topic in immunotherapy. Currently macrophage-centered therapy includes elimination of TAMs and re-polarization of TAMs into proinflammatory M1-type macrophages. Although the clinically beneficial for OS patients and effective therapy development has improved in recent years, the use of drugs that can change the M1/M2 ratio in favor of M1 (e.g., mifamurtide) require additional investigation [146].

In summary, the identification of new molecular targets, capable of regulating the polarization and recruitment of TAMs, is required if we are to improve the effectiveness of current OS therapies. Recently, it has been observed that macrophage PI 3-kinase γ [166], ERK5-MAPK [41] and cMaf [167] promote M2-like macrophage-polarization and control the switch between immune stimulation and suppression in cancer. Indeed, by blocking some of these signaling pathways it is possible to redirects TAM-polarization in favor of an M1 proinflammatory phenotype with both antitumor and anti-metastatic effects. Recently developed nanoparticle-based therapies provide potential platforms for tumor therapy advancement [168,169]. The use of nanoparticles loaded with specific intracellular-signaling molecular inhibitors, such as PI 3-kinase γ inhibitor [170], to change the TAM phenotype could be a promising alternative for the direct targeting of TAM therapy in OS.

It has been also been reported that cytotoxic CD8^+^ T lymphocytes are less abundant than myeloid cells in OS biopsies, suggesting that OSs are poorly immunogenic tumors [171,172]. Additional studies looking at the use of a chimeric antigen receptor (CAR) T cell therapy resulted in the development of CAR macrophages CAR-Ms [173]. CAR-Ms were shown to express proinflammatory cytokines and chemokines and were also noted to be effective in converting by stander M2 into M1 macrophages that demonstrated antigen-specific phagocytosis. Notably, a single infusion of human CAR-Ms decreased the tumor burden and prolonged overall survival in using a xenograft mouse model. In a humanized immune system (HIS) mouse model CAR-Ms were further shown to induce a proinflammatory TME response and to boost antitumor T cell activity. CAR macrophages are known to kill tumors both in human samples and in mouse models [173]. With genetically engineered macrophages, it may prove possible to directly influence the effectiveness of current combined therapies that are in use in OS patients beneficially.

## Figures and Tables

**Figure 1 ijms-21-05207-f001:**
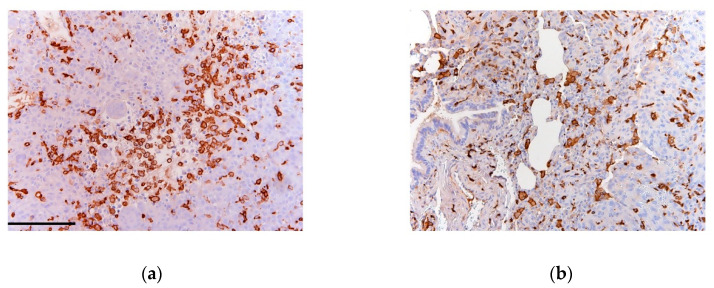
(**a**) Primary and (**b**; lung) metastatic osteosarcoma (OS) sections stained for CD163 (clone 10D6 Thermo Scientific). Dense infiltration of tumor-associated macrophages (TAMs) is indicated by diffuse stromal reactivity of CD163 (brown) as previously described [41].

**Figure 2 ijms-21-05207-f002:**
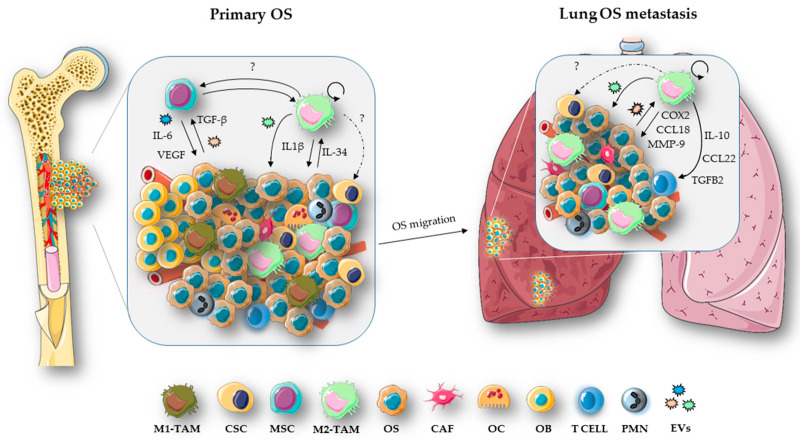
Role of TAMs in OS progression. Crosstalk between OS and stromal cells—via mechanisms such as activation of signaling pathways or exchange of extracellular vesicles (EVs), makes the tumor microenvironment (TME) conducive for cancer growth. In this regard, OS cells, by releasing TGF-β or EVs-delivered, support the malignant phenotype of the MSCs that in turn secrete cytokines such as IL-6 and VEGF molecules involved in promoting angiogenesis and metastasis [133]. A heterogeneous population of TAMs, with both M1 and M2 phenotypes, infiltrates the OS tumor [4]. M2-like TAMs play a key role in tumor progression and metastasis [133]. In particular, M2-like TAMs, by releasing IL-1β, induce chemoresistance in primary OS cells [96]. In response to the inflammatory stimuli, OS cells over express IL-34 that induces M2-like TAM recruitment, vasculogenesis and OS growth [7]. M2 macrophages are also involved in the maintenance of the stem-like phenotype of CSCs [99]. However, the exact mechanism by which TAMs and CSCs communicate in OS remains to be fully investigated, together with the possibility of a bidirectional link between TAMs and MSCs, as observed in other tumor types [124,125]. High numbers of TAMs in the TME are associated with metastatic OS tissue and poor survival rates for OS patients [134]. TAMs support tumor invasiveness via the secretion of CCL18 [98], proinflammatory molecules such as COX2 that, in turn, activate the IL-6/STAT3 pathway in OS cells and MMP-9, increasing the metastatic potential of the tumor [5]. OS-derived EVs have been shown to increase the expression of M2-type markers such as TGFB2, IL-10 and CCL22 in lung OS metastasis [101]. In addition, it has also been reported that proliferative TAMs support cancer metastasis process [135].

**Table 1 ijms-21-05207-t001:** Biomarkers associated with tumor-associated macrophages in OS.

Biomarkers	Function	Ref.
MMP-9	Matrix metalloproteinase	[5]
COX2	Proinflammatory enzyme	[5]
STAT3	Transcription factor	[5]
CD163	Scavenger receptor hemoglobulin	[6]
CCL18	Chemokine	[98]
CD209	Leptin receptor	[99]
IL-6	Interleukin	[100]
CCL22	Chemokine	[101]
IL-10	Interleukin	[101]
TGFB2	Cytokine	[101]
CD206	Mannose receptor	[102]

**Table 2 ijms-21-05207-t002:** Therapeutic agents targeting TAMs for OS treatment.

Agent	Mechanism	Phase	Ref.
All trans retinoic acid	Reduces polarization of M2-like	Pre-clinical	[8]
Mifamurtide Esculetin	Induces M1-like activation Inhibits TAMs differentiation	3 Pre-clinical	[145] [150]
Zoledronate	Polarizes TAMs to M1-like	3	[151]
Natalizumab	Interferes cross-talk between cancer cells and TAMs	NCT03811886	[157]
Nivolumab	mAbs anti-PD-1	NCT02304458	[158]
Pembrolizumab	mAbs anti-PD-1	NCT02301039	[158]

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
