# Peer review of "Tumor-Associated Macrophages in Osteosarcoma: From Mechanisms to Therapy"

_ijms, 2020, doi:10.3390/ijms21155207_

Round 1
Reviewer 1 Report
Reviewer comments
The manuscript entitles "Tumor-Associated macrophages in Osteosarcoma: from mechanisms to therapy" by Francesca Cersosimo et al. is a very interesting review on the different implications of macrophagic sub-populations in the osteosarcoma growth, resistances to treatments and metastatic dissemination. The reviewer has only one comment to this complete review which is the requirement of an edition of the text for English usage. Orthographic and grammatical mistakes present in the text have to be corrected before publication (for instance line 107 "by copy" not "be copy"; line 127 "tumorigenesis" not "tumorogenesis"...).
Author Response
The manuscript entitles "Tumor-Associated macrophages in Osteosarcoma: from mechanisms to therapy" by Francesca Cersosimo et al. is a very interesting review on the different implications of macrophagic sub-populations in the osteosarcoma growth, resistances to treatments and metastatic dissemination. The reviewer has only one comment to this complete review which is the requirement of an edition of the text for English usage. Orthographic and grammatical mistakes present in the text have to be corrected before publication (for instance line 107 "by copy" not "be copy"; line 127 "tumorigenesis" not "tumorogenesis"...).
Response:We apologize for these errors and have corrected. In addition, as you suggested, an extensive english editing and style revision of our manuscript was performed (all changes in the text are highlighted in the revised version of the manuscript).

Reviewer 2 Report
The review article title “Tumor-Associated Macrophages in Osteosarcoma: from mechanisms to therapy” In this review authors aim to give a current and comprehensive update about the known factors participating in the recruitment and activation of tumor-associated macrophages (TAMs) and the mechanism by which TAMs supports tumor metastasis in osteosarcoma (OS), focusing attention on the relationships between the signaling regulating the cross-talk between TAMs, cancer stem cells (CSCs), and mesenchymal stem cells (MSCs) in primary and metastatic OS. This article was well written. It was convincing. I prefer this article is suitable for publication but after some minor revision.
- Could the authors please provide separate tables for prognostic markers and therapeutic agents.
Author Response
The review article title “Tumor-Associated Macrophages in Osteosarcoma: from mechanisms to therapy” In this review authors aim to give a current and comprehensive update about the known factors participating in the recruitment and activation of tumor-associated macrophages (TAMs) and the mechanism by which TAMs supports tumor metastasis in osteosarcoma (OS), focusing attention on the relationships between the signaling regulating the cross-talk between TAMs, cancer stem cells (CSCs), and mesenchymal stem cells (MSCs) in primary and metastatic OS. This article was well written. It was convincing. I prefer this article is suitable for publication but after some minor revision.
1. Could the authors please provide separate tables for prognostic markers and therapeutic agents.
Response: We appreciate your comments and the assessment of our review manuscript.
As suggested, table 1 (Biomarkers associated with tumor-associated macrophages in OS) and table 2 (Therapeutic agents targeting TAMs for OS treatment) were added in the revised version of the manuscript at line 268 and line 445, respectively.
